# Characterization of Immune Response Diversity in Rodents Vaccinated with a Vesicular Stomatitis Virus Vectored COVID-19 Vaccine

**DOI:** 10.3390/v14061127

**Published:** 2022-05-24

**Authors:** Shen Wang, Cheng Zhang, Bo Liang, Weiqi Wang, Na Feng, Yongkun Zhao, Tiecheng Wang, Zhendong Guo, Feihu Yan, Songtao Yang, Xianzhu Xia

**Affiliations:** 1Key Laboratory of Jilin Province for Zoonosis Prevention and Control, Changchun Veterinary Research Institute, Chinese Academy of Agricultural Sciences, Changchun 130122, China; 18203762077@163.com (S.W.); zc1349@foxmail.com (C.Z.); lb19980526@163.com (B.L.); wangweiqi0712@163.com (W.W.); fengna0308@126.com (N.F.); zhaoyongkun1976@126.com (Y.Z.); wgcha@163.com (T.W.); guozd@foxmail.com (Z.G.); xiaxzh@cae.cn (X.X.); 2College of Veterinary Medicine, Hebei Agricultural University, Baoding 071000, China; 3College of Veterinary Medicine, Jilin University, Changchun 130028, China

**Keywords:** COVID-19, recombinant vesicular stomatitis virus, immunization routes, neutralizing antibody, challenge, rodent, animal models

## Abstract

Severe acute respiratory syndrome coronavirus 2 (SARS-CoV-2) has emerged as the prime challenge facing public health safety since 2019. Correspondingly, coronavirus disease 2019 (COVID-19) vaccines have been developed and administered worldwide, varying in design strategies, delivery routes, immunogenicity and protective efficacy. Here, a replication-competent vesicular stomatitis virus (VSV) vectored recombinant COVID-19 vaccine was constructed and evaluated in BALB/c mice and Syrian golden hamsters. In BALB/c mice, intramuscular (i.m.) inoculation of recombinant vaccine induced significantly higher humoral immune response than that of the intranasal (i.n.) inoculation group. Analyses of cellular immunity revealed that a Th1-biased cellular immune response was induced in i.n. inoculation group while both Th1 and Th2 T cells were activated in i.m. inoculation group. In golden hamsters, i.n. inoculation of the recombinant vaccine triggered robust humoral immune response and conferred prominent protective efficacy post-SARS-CoV-2 challenge, indicating a better protective immunity in the i.n. inoculation group than that of the i.m. inoculation group. This study provides an effective i.n.-delivered recombinant COVID-19 vaccine candidate and elucidates a route-dependent manner of this vaccine candidate in two most frequently applied small animal models. Moreover, the golden hamster is presented as an economical and convenient small animal model that precisely reflects the immune response and protective efficacy induced by replication-competent COVID-19 vaccine candidates in other SARS-CoV-2 susceptible animals and human beings, especially in the exploration of i.n. immunization.

## 1. Introduction

In December 2019, an emerging severe respiratory disease was first reported, named COVID-19 [1,2]. COVID-19 is caused by SARS-CoV-2, a single-stranded positive sense enveloped RNA virus. As of 5 May 2022, COVID-19 has caused more than 512 million confirmed cases and more than 6.24 million deaths worldwide [3]. Vaccination and the establishment of herd immunity are the main countermeasures confronting COVID-19. As of 5 May 2022, a total of 1.15 billion vaccine doses have been administered globally [3]. Of these vaccines, inactivated vaccines, virus vectored vaccines, subunit vaccines and mRNA vaccines are the main types of COVID-19 approved vaccines [4]. According to phase III clinical trials, the protective efficacy of these approved vaccines ranges from 50% to 95% [5,6,7,8,9,10].

The majority of approved COVID-19 vaccines are delivered through i.m. injection [4]. In addition, several novel intranasal (i.n.) or aerosol-delivered mucosal COVID-19 vaccines have been developed based on adenovirus virus vectors [11,12,13], chimpanzee adenoviruses (Simian Ad-36) [12,13], influenza virus vectors, parainfluenza virus vectors [14], as well as VSV vectors [15], etc. Mucosal vaccines present a convenient and effective new generation COVID-19 vaccine that mediates the initial recognition of pathogenicity and initiates pathogen-specific immune responses [16]. The establishment of both mucosal immunity and adapted immunity via mucosal immunization may confer better protection compared with traditional vaccines. Additionally, booster vaccinations with i.n.-delivered vaccines provide improved immunogenicity after the prime with the approved vaccines [17]. Nevertheless, pre-clinical data are limited for these mucosal COVID-19 vaccines. Furthermore, the selection and application of established COVID-19 animal models in mucosal vaccines demands further investigation.

VSV is a single-strand negative sense RNA virus belonging to Rhabdoviridae [18,19]. The relatively simple genome, wide host phagocytosis and promising replication ability, high growth titer and a maximum capacity of 4.5 kb foreign gene render VSV a distinguished vaccine vector since the establishment of the VSV reverse genetic system [20,21]. In previous studies, when the glycoprotein gene (G) of VSV was replaced with the spike (S) gene of SARS-CoV-2, the recombinant virus exhibited neutralizing activities and characteristics of various antibodies and ACE2-Fc soluble decoy protein similar to those of real SARS-CoV-2 [22]. As vaccine candidates, two groups developed replication-competent rVSV vaccines expressing the S protein of SARS-CoV-2, which conferred protection in golden hamsters and BALB/c mice against SARS-CoV-2 challenge via i.m. and intraperitoneal inoculation, respectively [23,24]. Subsequently, i.n. inoculation proved to be a more effective delivery route for nonhuman privates than that of the i.m. route [15]. Nevertheless, the route-dependent manner of the recombinant VSV-vectored COVID-19 vaccine in rodent animal models has not been exhaustively elucidated. A small animal model that reflects the real situation of other SARS-CoV-2 susceptible animals and human beings is urgently needed.

In this study, the replication-competent VSV-vectored recombinant COVID-19 vaccine was constructed and evaluated in the two most frequently applied COVID-19 rodent models in a point-to-point manner. Horizontal comparison of the delivery routes of this vaccine candidate enriches the pre-clinical information of immunogenicity and protective efficacy of the recombinant vaccine. Meanwhile, it provides directions for animal model selection in replication-competent COVID-19 vaccine evaluation and for the accurate development of COVID-19 mucosal vaccines.

## 2. Materials and Methods

### 2.1. Ethics Statement

The rescue of recombinant viruses was performed under biosafety level 2 (BSL2) conditions. All rodent experiments were conducted strictly according to the guidance of the animal welfare and ethics committee of Changchun Veterinary Research Institute, Chinese Academy of Agricultural Sciences, (approval number: IACUC of AMMS-11-2020-020). SARS-CoV-2 isolate (BetaCov/Wuhan/AMMS01/2020) is preserved in biosafety level 3 laboratory of Changchun Veterinary Research Institute. SARS-CoV-2 challenge study was conducted under biosafety level 3 (BSL3) conditions.

### 2.2. Plasmids

A reverse genetic system of rVSV was designed, synthesized and constructed as described previously [20,21]. Briefly, based on the full-length positive sequence of the VSV Indiana strain (genebank ID: nc_001560.1), The T7 promoter and hammerhead ribozyme sequence were added to the 5′ beginning of the whole gene sequence. A removable transcription unit encoding enhanced green fluorescence (eGFP) (GenBank ID: hv192862.1) was added between the N and P genes of VSV genome. Restriction enzyme sites *AscI*, *SanDI* and *PvuI*, *Bsu36I* were added upstream and downstream of G gene, respectively. Hepatitis D ribozyme sequence was added at the 3’ end of the whole gene sequence. The full-length plasmid was constructed in pcDNA3.1(+) vector by whole gene synthesis, named p3.1-VSV-eGFP. Plasmid without eGFP encoding unit was named p3.1-VSV. Supporting plasmid encoding VSV N, P, L and G proteins were constructed in pcDNA3.1(+) vector, named p3.1-VSV-N, p3.1-VSV-P, p3.1-VSV-L and p3.1-VSV-G, respectively.

The S protein coding sequence of SARS-CoV-2 mink variant cluster 5 (GISAID ID: EPI_ISL_616695) was synthesized and inserted between the *AscI* and *PvuI* sites into p3.1-VSV-eGFP and p3.1-VSV. The resulting plasmids were named p3.1-VSVΔG-S-eGFP and p3.1-VSVΔG-S, with the VSV glycoprotein G coding sequence being replaced by that of the SARS-CoV-2 S gene.

### 2.3. Cells, Antibodies and Proteins

BSR-T7 cells (ATCC, CCL-10) and Vero E6 cells (ATCC, CRL-1586) were maintained in Dulbecco’s modified Eagle’s medium (DMEM) supplemented with 10% fetal bovine serum (FBS), 1% L-glutamine and 1% penicillin-streptomycin solution (P/S) at 37 °C with 5% CO_2_. Rabbit polyclonal antibody against SARS-CoV-2 S protein (Cat. 40589-T62) was purchased from Sino Biological Inc (Beijing, China). The receptor binding domain (RBD) protein of SARS-CoV-2 (GISAID ID: EPI_ISL_616695) was produced by eukaryotic expression and purification.

### 2.4. Rescue and Identification of Recombinant VSV

The rVSVs were rescued by a reverse genetics approach. Briefly, BSR-T7 cells were transfected with pVSV plasmids and supporting plasmid encoding N, P, L and G of VSV using a calcium phosphate method (Thermo Fisher Scientific, Waltham, MA, USA) according to the manufacturer’s instruction (0.75 μg of p3.1-VSV-N, 1.25 μg of p3.1-VSV-P, 0.25 μg of p3.1-VSV-L, 2 μg of p3.1-VSV-G and 1.25 μg of the plasmid encoding one of the recombinant genomic clones described above). Sixty hours post-transfection, rVSVs in the supernatants were collected and stored at −80 °C. Serial passages were conducted in Vero E6 cells. The recombinant viruses were named rVSVΔG-S and rVSVΔG-S-eGFP, respectively. Recombinant VSV was identified through Western blotting and indirect immunofluorescence. Vero E6 cells with 80% confluent were infected with rVSVs at a multiplicity of infection (MOI) of 0.1. Following virus adsorption for 1 h at 37 °C, the inoculum was replaced with DMEM containing 5% FBS.

#### 2.4.1. Western Blot

rVSV-infected cell lysates were separated by 8% SDS-PAGE and immunoblotted with anti-S polyclonal antibody for 1 h at room temperature against SARS-CoV-2 S protein at a 1:2500 dilution. Following three wishes with PBST (phosphate-buffered saline containing 0.05% Tween-20), the samples were incubated with the HRP-conjugated anti-species-specific antibody (Bioword, Minnesota, MN, USA) at a 1:25,000 dilution. After another three washes with PBST, the samples were examined with Tanon 5200 chemiluminescence imaging system. Parental VSV served as a negative control (Same abbreviation in subsequent identifications).

#### 2.4.2. Indirect Immunofluorescent Staining

Cells were fixed 36 h post-infection with cold acetone. After inactivation, the cells were washed three times with PBST and incubated for 1 h at room temperature with the appropriate S protein-specific antibody diluted in phosphate-buffered saline (PBS). The samples were washed three times with PBST and incubated for another hour with an Alexa Fluor 568-conjugated anti-species-specific antibody (Thermo Fisher Scientific, Waltham, MA, USA). Then, nuclei were stained with appropriate diluted DAPI in PBS for 10 min. After being washed three times with PBST, the samples were examined with a Zeiss microscope.

#### 2.4.3. Virus Growth Curve

Vero E6 cells were infected with rVSVs at an MOI of 0.01. Supernatants were collected at the indicated time points post-infection and tittered by TCID_50_ using the Reed–Muench method.

### 2.5. Animal Immunization and Challenge

Six-week-old female BALB/c mice and four-week-old female Syrian golden hamsters (Mesocricetus auratus) were purchased from Beijing Vital River Laboratory Animal Technology Co., Ltd. (Beijing, China). On day 0 and day 21, BALB/c mice or golden hamsters were inoculated with rVSV-ΔG-S through i.m. or i.n. inoculation at a dose of 10^6^ TCID_50_/animal, 10 animals per group. Animals inoculated with the same volume of PBS through i.m. or i.n. inoculation served as the PBS inoculation control, 5 animals per group. Virus-specific antibodies, antibody subtypes and neutralizing antibodies were detected on day 14, 28 and 42 post-vaccination (dpv). In mice, IL-4 and IFN-γ in splenocytes were detected by ELISpot at 28 dpv. CD3^+^CD4^+^ and CD3^+^CD8^+^ T cells were detected by flow cytometry at 28 dpv. Golden hamsters were challenged with 10^5^ TCID_50_ SARS-CoV-2 at 35 dpv (10 animals from each vaccinated group and 4 animals from the PBS inoculation control). Challenged animals who were vaccinated with PBS were defined as challenged control (*n* = 4). Unchallenged animals who were vaccinated with PBS were defined as unchallenged control (*n* = 6). Clinical symptoms and weight of these challenged animals were recorded continuously within one-week post-challenge. Three days post-infection (dpi), turbinate and lung tissues were taken for TCID_50_ titration and RNA copy detection (5 animals from each vaccinated group and 2 animals from the PBS inoculation control). Meanwhile, pathological sections of lung tissue were analyzed by HE staining and immunohistochemistry (IHC).

### 2.6. ELISA and ELISpot

Enzyme linked immunosorbent assay (ELISA) was conducted according to a previous protocol [25]. First, 96-well microtiter plates (Corning-Costar, Corning, NY, USA) were coated overnight at 4 °C with recombinant RBD protein at 1 µg/mL. Following 3 washes with PBST and blocking for 2 h at 37 °C with PBS containing 3% BSA, the plates were incubated with 1:80–1:163,840 dilutions of serum samples in PBS containing 0.5% (*w*/*v*) BSA at 37 °C for 1 h. After another 3 PBST washes, the plates were incubated at 37 °C for 1 h with the following HRP-labeled appropriate diluted goat anti-mouse or goat anti-hamster IgG antibodies (Bioword, Minnesota, MN, USA). After the final 3 washes, 100 µL tetramethylbenzidine (TMB) substrate was added to each well and the color development was stopped with 50 µL/well H_2_SO_4_ for plate reading at 450 nm (Bio-Rad, Hercules, CA, USA).

Enzyme linked immunospot (ELISpot) was conducted according to the manufacturer’s instructions (Mabtech, 3311-4HPW-2, 3321-4HPW-2, Nacka Strand, Sweden). Splenocytes of mice were cultured in Dulbecco’s modified Eagle’s medium (DMEM; GIBCO, Grand Island, NY, USA) supplemented with 5% FBS (Gbico, Grand Island, NY, USA) and 1% penicillin-streptomycin (Sigma, St. Louis, MO, USA) and stimulated with 10 μg/mL RBD protein for 36 h at 37 °C, 5% CO_2_.

### 2.7. Flow Cytometry

Splenocytes of mice were isolated according to a previous protocol [25]. After 3 washes with PBS containing 0.5% FBS and blocking with CD16/CD32 for 30 min at 4 °C, splenocytes were stained with APC-CY7-labled CD3e, FITC-labled CD4 and PE-CY7-labeled CD8a antibodies (eBioscience, San Diego, CA, USA). Antibodies were diluted according to the manufacturer’s instructions. For the post-stimulated group, splenocytes were stimulated with 20 μg/mL RBD protein of SARS-CoV-2 for 48 h at 37 °C, 5% CO_2_ before staining. Data were collected from Backman Cytoflex flow cytometry.

### 2.8. nAbs Test

Serum samples of mice and golden hamsters were used for neutralizing antibody (nAbs) detection. All sera were heat-inactivated at 56 °C for 30 min, then diluted in 96-well plates with two-fold serial dilutions (from 1:20 to 1:40,960), mixed with 100 TCID_50_ of rVSVΔG-S-GFP and incubated at 37 °C, 5% CO_2_ for 1 h. Vero E6 cells were added in the virus-serum mixture in a volume of 50 µL/well. The 96-well plates were incubated at 37 °C, 5% CO_2_ for 48 h. The nAbs titer of each sample was the reciprocal of the serum dilution that eliminates all fluorescence signal.

### 2.9. Extraction of Viral RNA and Quantitative RT-PCR

Extraction of viral RNA and quantitative RT-PCR were conducted according to a previous protocol [25]. Briefly, tissue homogenates of the nasal turbinates and the lungs were prepared in an electric homogenizer for 300 s in 500 uL DMEM. The supernatant was collected and centrifuged at 12,000 rpm for 10 min at 4 °C. Viral RNA was extracted using the TIANGEN viral RNA Mini Kit (TIANGEN, Beijing, China) according to the manufacturer’s protocol. Viral RNA quantification was performed by quantitative reverse transcription PCR (qRT-PCR) targeting the N gene of SARS-CoV-2. qRT-PCR was performed with Premix Ex Taq (Takara, China) with the following primers and probes: NF (5′-GGGGAACTTCTCCTGCTAGAAT-3′); NR (5′-CAGACATTTTGCTCTCAAGCTG-3′); and NP (5′-FAM-TTGCTGCTGCTTGACAGATT-TAMRA-3′).

### 2.10. Quantification of Viral Loads by TCID50

Nasal turbinates and lung homogenate supernatants were serially diluted in DMEM and added to Vero E6 cells with 90% confluence in 96-well plates. The plates were incubated for 1 h at 37 °C with 5% CO_2_, and the inoculation was replaced with DMEM containing 2% FBS and 1% penicillin-streptomycin. After incubating for 72 h, the median tissue culture infective dose (TCID50) was detected by the cytopathic effect (CPE).

### 2.11. Histology and Immunohistochemistry (IHC)

The lungs were fixed in 4% paraformaldehyde, and paraffin sections were prepared routinely at 5 µm. The sections were stained with hematoxylin and eosin (H&E) for histopathologic examination.

For IHC, paraffin-embedded tissues were quenched for 30 min in aqueous 3% hydrogen peroxide. Following 3 washes with PBS, rabbit ployclonal anti-SARS-CoV-2 N antibody was applied to the section as the primary antibody (Sino Biological Inc, Beijing, China), at a 1:5000 dilution for 30 min. After another 3 washes with PBS, sections were incubated with the HRP-conjugated secondary antibody for 20 min at room temperature. After washing, diaminobenzidine (DAB) chromogenic solution was added. The positive region exhibits a brownish yellow color, and the color development is terminated by washing the section with tap water. Finally, the nucleus was stained with DAPI staining solution.

### 2.12. Statistical Analyses

GraphPad Prism 8.0. software (GraphPad Software Inc, San Diego, CA, USA) was used to analyze the data, which are expressed as the mean ± standard error of the mean (*SEM*). Significant differences between groups were determined using one-way ANOVA analysis of variance. *p* < 0.05 was considered statistically significant. Significance levels were defined as * *p* < 0.05, ***p* < 0.01, *** *p* < 0.001 and **** *p* < 0.0001.

## 3. Results

### 3.1. Characterization of rVSV-ΔG-S

Here, two replication-competent recombinant VSV viruses expressing the S gene of SARS-CoV-2 were developed (Figure 1a). According to the results of indirect immunofluorescence, rVSV-ΔG-S and rVSV-ΔG-S-eGFP were recognized by SARS-CoV-2-specific antibodies and presented red fluorescence (Figure 1b). A band representing S at 190 kDa was detected from infected cell lysates (Figure 1c). To assess the growth kinetics of rVSVs, viruses in the supernatant were measured every 12 h by TCID50. The peak titer of rVSV-ΔG-S and rVSV-ΔG-S-eGFP was 10^6.8^TCID50/mL at 72 h post-infection (Figure 1d). rVSV-ΔG-S served as a recombinant vaccine while rVSV-ΔG-S-eGFP was applied in the SARS-CoV-2 pseudovirus neutralization assay.

### 3.2. Immunogenicity of rVSV-ΔG-S in BALB/c Mice and Golden Hamsters

#### 3.2.1. Virus-Specific Antibody and Neutralizing Antibody

Wild-type BALB/c mice and golden hamsters were immunized with rVSV-ΔG-S via the i.m. or i.n. route at a dose of 10^6^TCID50. Body weight was monitored for one week and no obvious loss was observed in any of the groups (Figure 2a,d).

Neutralization antibody and virus-specific antibody elicited by the rVSV-ΔG-S via i.m. route were significantly higher than those of the i.n. route in BALB/c mice. The geometric mean titers (GMTs) of serum nAbs were 143 (reciprocal serum titer, same below) in the i.m. inoculation group at 56 dpv, significantly higher than that of the i.n. inoculation group which was 13 (*p* < 0.001) (Figure 2b). The GMT of the virus-specific antibody was 25,267 at 42 dpv, significantly higher than that of i.n. inoculation group, which was 3158 (*p* < 0.0001) (Figure 2c). However, in golden hamsters, i.n. immunization induced robust nAbs and virus-specific antibodies, significantly higher than those in the i.m. inoculation group. The GMT of serum nAbs and virus-specific antibody of the i.n. group were 1878 and 96,647, respectively, at 28 days, compared with that of 45 and 8640, respectively in the i.m. group (Figure 2e,f). The above results suggest a route-dependent manner of humoral immune response induced by recombinant vaccine in BALB/c mice and golden hamsters.

#### 3.2.2. Differential Immune Response Based on Inoculation Routes

To investigate the differential immune response induced by the rVSV vaccine in mice, virus-specific antibody subtypes IgG1, IgG2a and typical spleen cytokines including IL-4 and IFN-γ were evaluated. Furthermore, CD3+CD4+ and CD3+CD8+ positive T cells were investigated. The ratio of IgG2a/IgG1 was <1 in the i.m. inoculation group, while the ratio of IgG2a/IgG1 was >1 in the i.n. group (Figure 3a,d). The number of IFN-γ positive cell spots in the i.m. inoculation group was significantly higher than that in the i.n. inoculation group (*p* < 0.05). A certain amount of increased IFN-γ cell spots were observed in the i.n. inoculation group compared with the PBS inoculation control (Figure 3b). The number of IL-4 cell spots in the i.m. inoculation group was significantly higher than that in the i.n. inoculation group and PBS inoculation control group (*p* < 0.001) (Figure 3c), no difference was observed between the i.n. inoculation group and the PBS inoculation control. The above results suggest that the recombinant vaccine could promote IFN-γ and IL-4 splenocytes via i.m inoculation, while IFN-γ splenic lymphocytes were promoted via i.n inoculation. Accordingly, significantly higher CD3+CD4+ and CD3+CD8+ positive T cells were observed in the i.m. group whilst significantly higher CD3+CD8+ positive T cell were observed in i.n. group (*p* < 0.05) (Figure 3e,f). Taken together, the above results indicate that a Th1-biased immune response was induced via i.n. inoculation of the recombinant vaccine whilst both Th1 and Th2 cells were activated via i.m. inoculation.

### 3.3. rVSV-ΔG-S Protects Golden Hamsters against SARS-CoV-2 Challenge

Humoral immune response elicited by rVSV-ΔG-S via the i.n. route was significantly higher than that via the i.m. route in golden hamsters. To assess whether this vaccine candidate delivered by i.n. can confer better protection against SARS-CoV-2, groups of golden hamsters from the i.n. inoculation group (*n* = 10), i.m. inoculation group (*n* = 10) and PBS inoculation group (*n* = 4) were i.n. challenged with 10^5^TCID_50_ SARS-CoV-2 at 42 dpv after two doses of rVSV-ΔG-S immunization. Body weight was monitored for a week post-challenge. The i.n. group showed a slight weight increase of 16.3% from the initial weights of the hamsters prior to the challenge, which was comparable to that of the unchallenged control group (*n* = 6) who were vaccinated with PBS (Figure 4a). The i.m. group displayed a weight increase of 5.3% at 7 dpi (Figure 4a). The weight gain was stalled in the challenged control group who were vaccinated with PBS (*n* = 4). The golden hamsters were sacrificed for the viral load test and pathogenesis analysis at 3 dpi.

The lung viral titers and RNA load were high in challenged control animals. Viral RNA levels of the i.n. inoculation groups were 6.8 and 1.6 logs lower than the PBS inoculation control group in the lungs and nasal turbinates, respectively (*p* < 0.0001, *p* < 0.05). Viral RNA levels in the lungs were under the limit of detection (Figure 4c–f). For viral titers, i.n. inoculation groups were 5.4 and 4.8 logs lower than the PBS inoculation control group in the lungs and nasal turbinates, respectively (Figure 4c–f) (*p* < 0.0001, *p* < 0.001). In the i.m. group, viral RNA and viral titers in nasal turbinates were still at a high level although slightly lower than those in the PBS inoculation control. Viral RNA and viral titers in the lungs were 2 and 2.6 logs reduced compared to the PBS inoculation control (*p* < 0.0001, *p* < 0.0001).

Lung sections of challenged animals were stained with hematoxylin-eosin (H&E) and subjected to immunohistochemistry (IHC) (Figure 5). H&E staining showed that the lung tissue structure of the challenged PBS inoculation group was moderately abnormal, thickened alveolar wall, proliferated epithelial cells (yellow arrow), broken alveolar diaphragms and atrophic alveoli (red arrow) were observed. In the i.m. inoculation group, the lung tissue structure was moderately abnormal, damaged alveolar structure and thickened local alveolar wall were observed (yellow arrow). The nuclei of alveolar epithelial cells were clear. No inflammatory cell infiltration was found in the lung parenchyma. In the i.n. inoculation group and the unchallenged control group, the lung tissue structure was normal, the alveolar wall in the visual field was uniform and there was no thickening, the alveolar outline was clear and the size was normal, the alveolar epithelial nucleus was round without necrosis. No inflammatory cell infiltration was found in the lung parenchyma. Immunostaining of SARS-CoV-2 antigen showed that i.n. inoculation significantly reduced the viral infection in the lungs, while a large amount of SARS-CoV-2 infection could be detected in the lung tissues of animals in the i.m. immunization group and PBS inoculation control group (red arrow).

These results suggest that rVSV-ΔG-S generated protective immunity, which limited SARS-CoV-2 replication in the lungs and nasal turbinates. The protective efficacy in the i.n. group was better than that of the i.m. group.

## 4. Discussion

Containing the outbreak and spread of COVID-19 depends significantly on the development and delivery of COVID-19 vaccines. As of 28 April 2022, 153 COVID-19 vaccine candidates are in clinical development whilst 196 vaccine candidates are in pre-clinical development [4]. For pre-clinical evaluation of COVID-19 vaccines, BALB/c mice and golden hamsters are established economic and convenient small animal models [26,27,28,29,30,31,32,33,34,35,36]. Golden hamsters are naturally susceptible to SARS-CoV-2, while mice are insusceptible to the original strain SARS-CoV-2. When these rodent animal models are engaged in the evaluation of COVID-19 vaccines, few consensuses have been achieved in terms of the selection of appropriate animal models, especially in the evaluation of novel i.n.-delivered mucosal COVID-19 vaccine candidates.

In this study, the replication-competent VSV-vectored recombinant vaccine was evaluated in the rodent animal models discussed above. In BALB/c mice, the recombinant vaccine induced significantly higher nAbs and virus-specific antibodies via i.m. inoculation than that of i.n. inoculation. Antibody subtype analysis revealed that i.n. inoculation of the recombinant vaccine induced a safe Th1-biased immune response with low risk of antibody-dependent enhancement (ADE) in BALB/c mice [37]. Whereas in i.m. inoculation group, both Th1 and Th2 cellular immunity were activated. Nevertheless, an opposite result was observed in golden hamsters, that is, i.n. inoculation of the recombinant vaccine elicited robust nAb and virus-specific antibodies, which were significantly higher than those of the i.m. inoculation group. The SARS-CoV-2 challenge study in golden hamsters was confirmed to the immunogenicity data. I.n. route vaccination conferred a better protection in golden hamsters than that of i.m. route vaccination. In a previous study [15], similar phenomena have been presented, the nAb levels elicited via i.n. route of replication-competent VSV-vectored recombinant vaccine were approximately 8-fold higher than those via i.m. inoculation in nonhuman primates, which was opposite to the results in hACE2 knock-in mice models. Of note, in January 2021, Merck announced the suspension of the clinical trial of a i.m.-delivered VSV-based COVID-19 vaccine, stating that “the results were disappointing and somewhat unexpected”. The above clinical result may be related to the route of immunization, as has been described in golden hamsters. Taken together, in the evaluation of replication-competent COVID-19 vaccines, golden hamsters exhibit a high degree of consistency compared with nonhuman privates and human beings.

Differences in binding efficacy and the distribution of ACE2 may explain the route-dependent manner of the recombinant vaccine in BALB/c mice and golden hamsters. According to the binding ability of the SARS-CoV-2 spike protein to different ACE2 orthologs from different origins, murine ACE2 exhibits limited binding ability while hamster ACE2 shows potent binding efficacy [38,39]. Furthermore, the distribution and abundance of ACE2 varies in different organs, which may be correlated with the difference in immunogenicity between inoculation routes. ACE2 is broadly expressed in the upper respiratory tract and lungs while the distribution of ACE2 is low in skeletal muscle [40,41]. In terms of the binding ability of ACE2, the susceptibility to SARS-CoV-2 as well as the consistency in immune response with nonhuman primates, golden hamsters may represent a more feasible small animal for replication-competent vaccine candidates.

The differentiation direction of Th0 cells depends on the antigen and cytokines in the local environment. For the replication-competent VSV-vectored vaccine, i.m. inoculation accomplished the systemic distribution of antigen, which may contribute to the activation of both Th1 and Th2 T cells. For i.n. inoculation routes, the recombinant vaccine simulates the nature infection of SARS-CoV-2, Th1 cells were activated, similar to that of mild COVID-19 patients [42,43]. The limitation of this study is that SARS-CoV-2-specific T cellular immunity in golden hamsters was not detected, which further limits the comparison of T cell activation in these rodent models.

According to previous studies, nAbs plays a key role in the protection of the SARS-CoV-2 challenge [44,45,46]. In the SARS-CoV-2 challenge study of golden hamsters, we confirmed that nAbs was correlated with the protective efficacy. Animals with higher SARS-CoV-2 nAbs received better protection. Our data indicate that rVSVΔG-S may be suitable for i.n. inoculation instead of the traditional i.m. inoculation in other SARS-CoV-2 susceptible animal and human beings. Particularly, in the i.n. inoculation group, a better protection in upper respiratory was achieved in terms of the viral RNA and viral titer in the nasal turbinates. Unfortunately, sterile immunity was not achieved in the nasal turbinates. The residual of SARS-CoV-2 in the upper respiratory has been presented as an overall concern facing COVID-19 vaccine development [47], this characteristic has been fully recapitulated in golden hamsters. In a subsequent study, this issue should be further investigated.

In this study, a replication-competent VSV-vectored recombinant vaccine was constructed and evaluated in two rodent animal models. We confirmed that i.n. inoculation of the recombinant vaccine elicits robust humoral immunity and confers potent protective efficacy in golden hamsters post-SARS-CoV-2 challenge. Longitudinal comparison of the i.n. and i.m. delivery route of the recombinant vaccine helps to elucidate the route-dependent manner in two rodent animal models, and in particular, it emphasizes that the golden hamster may be an appropriate small animal for pre-clinical evaluation of these replication-competent and i.n.-delivered COVID-19 vaccine candidates.

## 5. Conclusions

The current study results showed that a replication-competent rVSV vaccine via the i.n. route and not the i.m. route effectively reduces SARS-CoV-2 infections in the lungs and confers protection against virus-induced lung pathology in golden hamsters, which is a promising experimental rodent model for testing COVID-19 vaccine candidates.

## Figures and Tables

**Figure 1 viruses-14-01127-f001:**
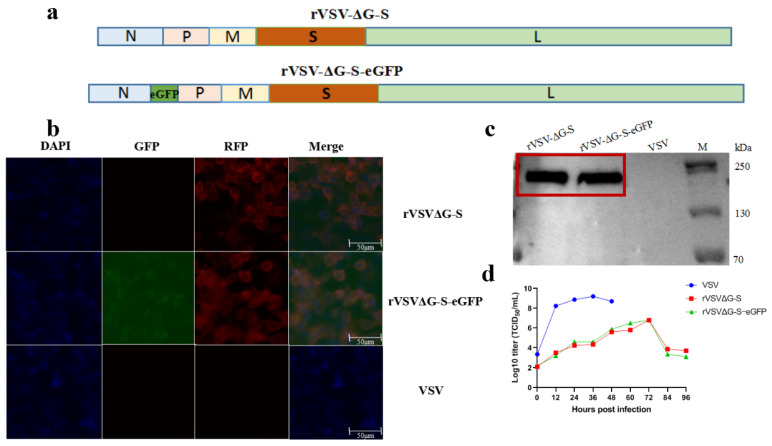
Characterization of the recombinant VSV viruses. (**a**) Schematic diagrams showing genome organization of the rVSV. VSV g was replaced by SARS-CoV-2 S to generate rVSV-ΔG-S and rVSV-ΔG-S-eGFP. (**b**) Indirect immunofluorescence identification results of rVSV-ΔG-S and rVSV-ΔG-S-eGFP. Vero E6 cells were infected with rVSV-ΔG-S or rVSV-ΔG-S-eGFP and recognized by SARS-CoV-2-specific antibody and Alexa Fluor 568-conjugated secondary antibody, presented red fluorescence, rVSV-ΔG-S-eGFP presented green fluorescence. No fluorescence signal was observed in the VSV control. (**c**) Lysates of rVSV-ΔG-S/rVSV-ΔG-S-eGFP-infected Vero E6 cells, and VSV-infected Vero cells were blotted with an antibody recognizing SARS-CoV-2 S protein, a band representing S at 190 kDa was detected in rVSV-ΔG-S and rVSV-ΔG-S-eGFP group, no band was observed in the VSV control. (**d**) Growth kinetics of rVSV-ΔG-S, rVSV-ΔG-S-eGFP and VSV. Vero E6 cells were infected with the recombinant VSV viruses (MOI = 0.01) and virus titers in the supernatant were measured at the indicated time points post-infection.

**Figure 2 viruses-14-01127-f002:**
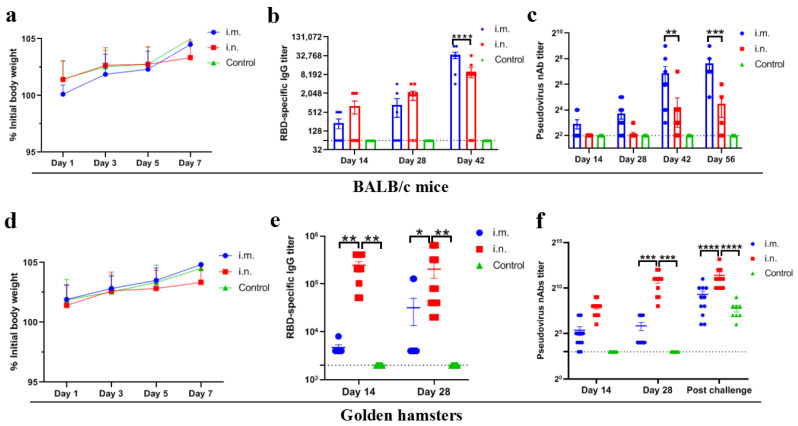
Virus-specific antibody and neutralizing antibody induced by rVSVΔG-S. (**a**) Weight change monitoring of BALB/c mice following vaccination for 7 days. i.m., i.n. and control refer to i.m. (*n* = 10), i.n. inoculation group (*n* = 10) and PBS inoculation control group (*n* = 10), respectively same in (**b**–**f**). (**b**) RBD-specific IgG in mouse serum samples at 14, 28 and 42 dpv. (**c**) nAbs in the serum of mice at 14, 28, 42 and 56 dpv. (**d**) Weight changes in golden hamsters 7 days after vaccination. (**e**) RBD-specific IgG in the serum of golden hamsters at 14 and 28 dpv. (**f**) nAb titers in hamster serum samples at 14, 28 dpv and post-SARS-CoV-2 challenge. Data are presented as mean ± SEM. (* *p* < 0.05, ** *p* < 0.01,*** *p* < 0.001, **** *p* < 0.0001).

**Figure 3 viruses-14-01127-f003:**
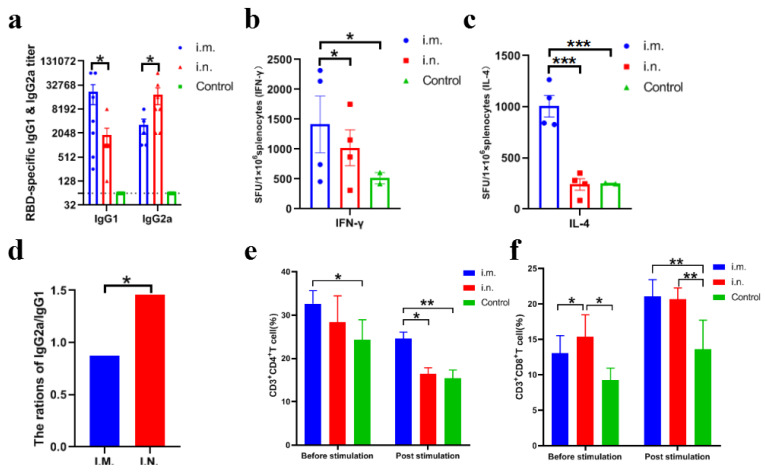
Antibody subtype and cytokines induced by rVSVΔG-S in BALB/c mice. I.m., i.n. and control refer to i.m. (*n* = 4), i.n. inoculation group (*n* = 4) and PBS inoculation control group (*n* = 2), respectively (same in Figure 3b–f). (**a**) RBD-specific IgG1 and IgG2a induced by the rVSV vaccine in mice. (**b**,**c**) Splenocyte IFN-γ and IL-4 induced by the rVSV vaccine in mice. (**d**) Ratio of IgG2a/IgG1 in i.m. and i.n. group. (**e**,**f**) CD3+CD4+ and CD3+CD8+ positive T cell proportions in splenocyte before and post-stimulation with 20 μg/mL RBD protein of SARS-CoV-2 for 36 h at 37 °C, 5% CO_2_. Data are presented as mean ± SEM. (* *p* < 0.05, ** *p* < 0.01, *** *p* < 0.001).

**Figure 4 viruses-14-01127-f004:**
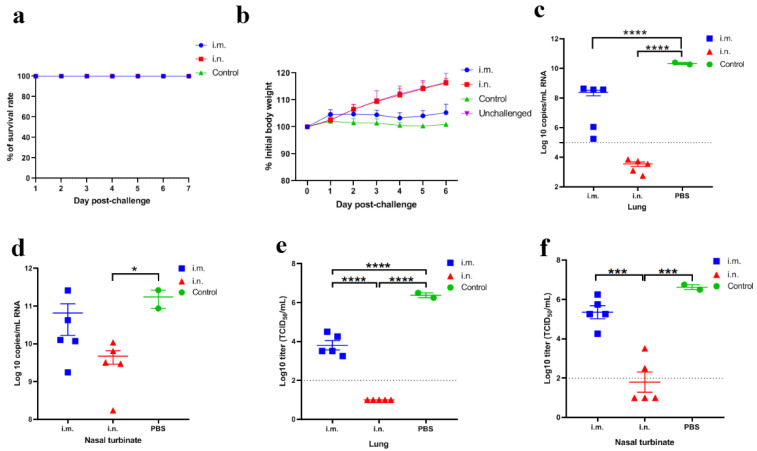
Protective efficacy of rVSV-ΔG-S in golden hamsters following SARS-CoV-2 challenge. i.m., i.n. and control refer to i.m. (*n* = 5), i.n. inoculation group (*n* = 5) and PBS inoculation group (*n* = 2) who were challenged with SARS-CoV-2, respectively (same in Figure 2b–f). Unchallenged control refers to hamsters inoculated with PBS and free from SARS-CoV-2 challenge (*n* = 6). (**a**) Survival rate following SARS-CoV-2 challenge. (**b**) Body weight change in golden hamsters following SARS-CoV-2 challenge. (**c**,**d**) SARS-CoV-2 viral RNA copies in the lungs and nasal turbinates following SARS-CoV-2 challenge. (**e**,**f**) SARS-CoV-2 viral titers in the lungs and nasal turbinates following SARS-CoV-2 challenge. Data are presented as mean ± SEM. (* *p* < 0.05, *** *p* < 0.001, **** *p* < 0.0001).

**Figure 5 viruses-14-01127-f005:**
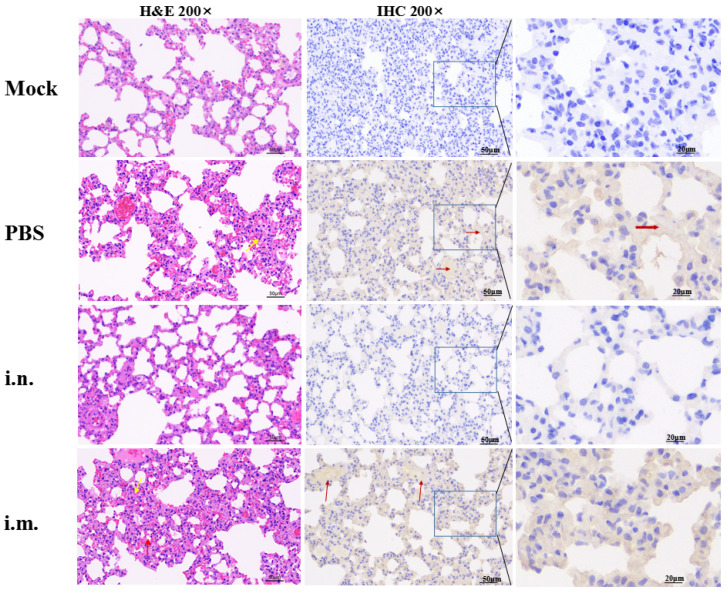
Representative H&E staining and IHC of lung sections from SARS-CoV-2-infected golden hamsters at 3 days post-infection. Mock refers to hamsters inoculated with PBS and free from SARS-CoV-2 challenged. i.m., i.n. and control refer to i.m. inoculation group, i.n. inoculation group and PBS inoculation group who were challenged with SARS-CoV-2, respectively. In H&E staining sections, the thickened alveolar wall and proliferated epithelial cells are marked with yellow arrows, broken alveolar diaphragms and atrophic alveoli are marked with red arrows. In IHC sections, SARS-CoV-2 antigens that were recognized by SARS-CoV-2-N-specific antibodies were marked with red arrows.

## Data Availability

Not applicable.

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
