# Peer review of "Characterization of Immune Response Diversity in Rodents Vaccinated with a Vesicular Stomatitis Virus Vectored COVID-19 Vaccine"

_viruses, 2022, doi:10.3390/v14061127_

Round 1
Reviewer 1 Report
Shen Wang and col. present an interesting study on a vaccine against SARS-COV 2 based on the vesicular stomatitis virus and the SARS-cov2 spike protein. Although the vaccine approach is not new, they explore intranasal immunization and compare it with intramuscular immunization in BalB/C mice and golden hamsters. The results show that intranasal immunization induces a powerful humoral and cellular response of Th1 character, with respect to the intramuscular route (Th1/Th2). Being also protective against the viral challenge. Another important finding of the work is the susceptibility of golden hamsters to SARS-COV2 infection, something that does not occur in wild mice. Proposing itself as a good model for testing new vaccines.
Mayor comments:
1) In intranasal immunizations, could the authors measure the presence of IgA in the serum of animals, and how is it compared to intramuscular ones?
Minor comments:
1) The authors should improve the quality of the image in Figure 1b: Indirect immunofluorescence identification results of rVSV-ΔG-S/rVSV-ΔG-S-eGFP.
2) In figure 5, they should indicate what the arrows point to in the images .
Author Response
Please see the attachment, thank you.

Reviewer 2 Report
Please see the attachment for specific comments.

Author Response
Please see the attachment, thank you.
